# A New Paradigm on Waste-to-Energy Applying Hydrovoltaic Energy Harvesting Technology to Face Masks

**DOI:** 10.3390/polym16172515

**Published:** 2024-09-04

**Authors:** Yongbum Kwon, Dai Bui-Vinh, Seung-Hwan Lee, So Hyun Baek, Hyun-Woo Lee, Jeungjai Yun, Inhee Cho, Jeonghoon Lee, Mi Hye Lee, Handol Lee, Da-Woon Jeong

**Affiliations:** 1Korea National Institute of Rare Metals, Korea Institute of Industrial Technology, Incheon 21655, Republic of Korea; kyb916@kitech.re.kr (Y.K.); leesh93@kitech.re.kr (S.-H.L.); qorthgus9@kitech.re.kr (S.H.B.); totptkd12@kitech.re.kr (H.-W.L.); yjj0011@kitech.re.kr (J.Y.); cdcih@kitech.re.kr (I.C.); lmh62@kitech.re.kr (M.H.L.); 2Department of Environmental Engineering, Inha University, Incheon 22212, Republic of Korea; daibv@ntu.edu.vn; 3Manufacturing AI Research Center, Korea Institute of Industrial Technology, Incheon 21999, Republic of Korea; kokonut@kitech.re.kr; 4Program in Environmental and Polymer Engineering, Graduate School of Inha University, Incheon 22212, Republic of Korea; 5Particle Pollution Research and Management Center, Incheon 21999, Republic of Korea

**Keywords:** energy harvesting, hydrovoltaic generator, carbon black coating, face mask recycling, waste to energy (WTE)

## Abstract

The widespread use of single-use face masks during the recent epidemic has led to significant environmental challenges due to waste pollution. This study explores an innovative approach to address this issue by repurposing discarded face masks for hydrovoltaic energy harvesting. By coating the face masks with carbon black (CB) to enhance their hydrophilic properties, we developed mask-based hydrovoltaic power generators (MHPGs). These MHPGs were evaluated for their hydrovoltaic performance, revealing that different mask configurations and sizes affect their efficiency. The study found that MHPGs with smaller, more structured areas exhibited better energy output, with maximum open-circuit voltages (*V_OC_*) reaching up to 0.39 V and short-circuit currents (*I_SC_*) up to 65.6 μA. The integration of CB improved water absorption and transport, enhancing the hydrovoltaic performance. More specifically, MHPG-1 to MHPG-4, which represented different sizes and features, presented mean *V_OC_* values of 0.32, 0.17, 0.19 and 0.05 V, as well as mean *I_SC_* values of 16.57, 15.59, 47.43 and 3.02 μA, respectively. The findings highlight the feasibility of utilizing discarded masks in energy harvesting systems, offering both environmental benefits and a novel method for renewable energy generation. Therefore, this work provides a new paradigm for waste-to-energy (WTE) technologies and inspires further research into the use of unconventional waste materials for energy production.

## 1. Introduction

An epidemic disease has drastically altered everyday life, leading to a significant increase in the use of personal protective equipment (PPE), especially face masks [1]. While face masks have been essential in curbing the spread of the virus, their widespread use has resulted in a substantial environmental challenge [2,3]. The surge in demand for single-use face masks has exacerbated plastic pollution, as these masks are predominantly composed of non-biodegradable materials such as polypropylene [4,5,6]. This has created a pressing need for innovative waste management solutions that not only mitigate environmental burdens but also repurpose waste into valuable resources.

Waste-to-energy (WTE) technologies have emerged as a promising avenue for addressing the dual challenges of waste management and renewable energy production. These technologies convert waste materials into usable energy forms, such as electricity, heat and alternative fuel [7]. Traditional WTE methods include incineration, anaerobic digestion and gasification, each with its own advantages and limitations [8,9]. However, the quest for more sustainable and efficient technologies continues, with recent advancements exploring novel materials and processes for energy harvesting. One such novel approach is hydrovoltaic energy harvesting, which harnesses the energy from evaporating water to generate electricity [10,11]. This method leverages the natural process of water evaporation, which is a continuous and abundant source of energy. By using different kinds of materials that facilitate the evaporation process, it is possible to create systems that convert this energy into electricity [12,13,14].

For example, Park et al. [15] present a novel approach to sustainable energy generation using a leaf-inspired energy-harvesting foam (LIEHF) composed of polydimethylsiloxane integrated with conductive carbon materials. The LIEHF can produce an open-circuit voltage (*V_OC_*) of 254 mV, a short-circuit current (*I_SC_*) of 42 μA and a power density of 1 μW cm^−2^ under standard sunlight conditions. A honeycomb-structured reduced graphene oxide (rGO) film was demonstrated to generate electricity well through water evaporation [16]. The innovative structure of the rGO film significantly enhanced the evaporation-induced output power, with its abundant interconnected microchannels providing a larger specific surface area and superior ion-exchange capacity. This generator was capable of stable performance over 240 h, delivering a stable *V_OC_* of approximately 0.83 V with a power density of 0.79 μW cm^−2^. Additionally, typical bio-based materials such as wood can be chemically treated to achieve a highly porous structure [17]. Despite the current limitation of maintaining high voltage for only 2–3 h, the wood power generator has the potential to produce up to 1.0 V and a power output of 1.35 μW cm^−2^. Porous carbon film has also demonstrated the significant role of carbon materials in evaporation-driven flow, generating a sustainable voltage of approximately 1.0 V with a power density of approximately 8.1 μW cm^−3^ under ambient conditions [18]. The authors have highlighted further application potentials of the device, such as electrodeposition, water splitting and pollutant degradation, making it low-cost and environmentally friendly.

In this work, we explored the potential of using discarded face masks as a key material in a hydrovoltaic energy harvesting system by investigating feasibility and efficiency. The choice of discarded face masks is both timely and innovative. To date, there has been limited research on the use of face masks in energy harvesting applications. Most studies have focused on the environmental impact of face mask waste or their potential to be recycled into new materials [19,20,21,22]. Given the sheer volume of face mask waste generated globally, repurposing these masks into an energy harvesting system addresses multiple issues simultaneously. It provides a method of reducing plastic pollution, offers a new source of renewable energy and adds value to what would otherwise be considered waste. Relatively few studies have explored the techno-economic aspects of hydrovoltaic energy harvesting systems, especially when compared to other energy conversion technologies such as triboelectric [23,24], piezoelectric [25,26,27] and other renewable energy generators [28,29,30]. From this perspective, the potential economic benefits of hydrovoltaic energy harvesting using face masks could also be a valuable focus for future research. Finally, repurposing masks aligns with the principles of a circular economy, where waste materials are reintegrated into the production cycle, minimizing waste and optimizing resource use. The significance of this research lies in its potential to provide a sustainable solution to a pressing environmental problem. By transforming discarded face masks into a source of renewable energy, this study not only addresses the issue of plastic pollution, but also contributes to the diversification of energy sources.

## 2. Materials and Methods

### 2.1. Materials

Single-use disposable face masks were obtained as commercial products. Carbon black powder (CB) was obtained from Mitsubishi Chemical Co., Ltd., Tokyo, Japan. To uniformly disperse the CB in deionized (DI) water, cetyl trimethyl ammonium bromide (CTAB), purchased from Tokyo Chemical Industry Co., Ltd., Tokyo, Japan, was used. To fill the nanochannels with the ion solution for electricity generation, lithium chloride (LiCl, ≥99.98%) powders were purchased from Sigma-Aldrich Co., Ltd., St. Louis, MO, USA. An adhesive copper foil tape for connecting electrodes to the generators was purchased from Teraoka Seisakusho Co., Ltd., Tokyo, Japan.

### 2.2. Preparation of Mask-Based Hydrovoltaic Power Generator (MHPG)

To prepare the CB coating solution, 5.0 g of CB was dispersed in 800 mL of DI water with 10.0 g of CTAB. The CB solution was then mixed via sonication for 60 min to uniformly disperse. To fabricate the mask-based hydrovoltaic power generators (MHPGs), single-use disposable face masks with the elastic ear loops and metal strips removed were dip-coated in the CB solution and then dried in a 90 °C oven (SH Scientific Co., Ltd., Sejong, Republic of Korea) for 2 h. After that, two pieces of adhesive copper foil tape were connected to each end of the CB-coated face mask and MHPGs were eventually prepared. To investigate the characterizations of MHPGs, scanning field emission electron microscopy (SEM) and energy-dispersive X-ray spectroscopy (EDS) (JSM-7100F, JEOL Co., Ltd., Tokyo, Japan), as well as Fourier-transform infrared spectroscopy (FT-IR, Spectrum Two, PerkinElmer Inc., Waltham, MA, USA), were utilized.

### 2.3. Electrical Output Performance by Different Features

Because hydrovoltaic generation performances are dependent on the generator heights, widths and thicknesses, different features and shapes of MHPGs were designated, as summarized in Table 1. A typical deliquescent chemical, 3 M of lithium chloride (LiCl) solution, was used as liquid source of hydrovoltaic energy harvesting. The generation performances, including open-circuit voltage and short-circuit current curves, were measured using a Keithley 2400 source meter (Keithley Instruments, Inc., Cleveland, OH, USA) at 23 °C and a relative humidity of 40%.

## 3. Results and Discussion

### 3.1. Characterizations of MHPGs

Single-use face masks were repurposed as MHPGs following the experimental process shown in Figure 1a. In this study, four MHPGs with differing shapes and features (i.e., MHPG-1, MHPG-2, MHPG-3 and MHPG-4) were examined, as illustrated in Figure 1b. The surface of the single-use face masks was inspected to verify its disordered fibrous structure, with fiber diameters ranging from 20.0 to 23.5 μm, as shown in the SEM image (Figure 1c). Due to the application of CTAB, a uniform distribution of CB on the mask surface was observed. As shown in Figure 1d, nano-sized CB powder aggregated on the polypropylene fibers, offering a larger specific surface area compared to the non-coated (raw) face mask. In the hydrovoltaic energy harvesting system, CB plays an important role in enhancing the hydrophilic characteristics within the microchannels of the face mask. With its abundant capillary structure and high surface-area-to-volume ratio, the face mask with CB permits better water absorption and transport abilities. Additionally, MHPGs exhibit good flexibility, as they can be repeatedly bent and stretched back to their original shape without cracking or crumbling (Figure 1e). Thus, the mechanical properties of MHPGs further confirm their reliability as an energy harvesting device.

The CB-coated mask often has a porous structure due to the inherent voids and gaps between the aggregated nanoparticles. This porosity is beneficial for hydrovoltaic applications as it enhances water absorption and transport, which are critical for efficient energy harvesting. The nanostructured coating provides a large surface area relative to its volume. This high surface area is crucial for maximizing interaction with water, allowing more effective capillary action and ion transport, leading to better hydrovoltaic performance (Figure 2a). The surface modification of the face mask was further confirmed by FT-IR spectra. As shown in Figure 2b, the absorption peak at 2950 cm^−1^, corresponding to the stretching vibration of carboxylic acid -OH, was significantly enhanced due to the CB coating process on the mask surface. This can contribute to increasing the hydrophilic property of MHPGs, ensuring continuous water transport and evaporation for energy harvesting. The absorption peak at 2917 cm^−1^ and the absorption peak at 2836 cm^−1^ on the curves of both the raw face mask and the MHPGs were respectively attributed to C-H alkane and C-H aldehyde. As signature peaks of polypropylene, the absorption peaks at 1453 cm^−1^ and 1375 cm^−1^, corresponding to the functional group of -CH_3_, could be seen on both curves. Additionally, there was an increasing absorption peak at 1167 cm^−1^ (C-O-C) attributed to carbon black powder on the surface of the MHPGs. The uniform distribution of CB on the MHPGs was clearly apparent in the EDS patterns examined through Figure 1c,d. As evidenced in Figure 2c, carbon (C, red) was evenly dispersed on the surface of the face mask at the nanometer level due to the polypropylene composition of the mask filter. Although the additional carbon (C, red) from the CB is indistinguishable from the polypropylene microfibers, the successful distribution of CB can be verified by the presence of bromine (Br, green) owing to the mixture of CTAB in the CB solution (Figure 2d).

### 3.2. Electrical Performances by Different Features

Following the general energy harvesting principle and generation mechanism, the bottom electrode of the MHPGs was immersed in liquid. In this study, a typical deliquescent chemical, the LiCl solution, was applied rather than DI water. There are mainly four principal mechanisms for hydrovoltaic conversion, including streaming potential, electron drag effect, pseudo-streaming mechanism and ion gradient diffusion, and those mechanisms are known to co-exist simultaneously [31]. When water is deliberately injected or absorbed from the moist air into the hydrovoltaic energy harvesting generator, electrokinetic phenomena occur due to the movement of charged ions through the liquid electrolyte flowing over the capillary channel. At that moment, ion solutions such as sodium chloride (NaCl), calcium chloride (CaCl_2_) and hydrochloric acid (HCl) can improve the electricity generation output better than DI water [32,33]. Similar to the roots of plants, the water absorbed from one end of the MHPG follows the capillary channel. Water evaporation then occurs at the other end of the MHPG, which generates electricity due to the potential difference between the wet and dry sides (Figure 3a). As demonstrated by Bae et al. [34], the continuous capillary flow of injected moisture is derived at the interface of the wet and dry sides of the generator. This asymmetrical wetness on the device and the water supply cycle are essential for generating electric power, causing streaming potential, a streaming current, an electro drag effect and ion gradient diffusion, which are the four principle mechanisms for hydrovoltaic conversion [31,35].

Comparing the electricity output performance of the four different generating devices (MHPG-1, MHPG-2, MHPG-3 and MHPG-4), the maximum *V_OC_* values were 0.39, 0.31, 0.28 and 0.28 V, respectively (Figure 3b), and the maximum *I_SC_* values were 18.3, 32.7, 65.6 and 17.3 μA, respectively (Figure 3c). Thus, the maximum power outputs of 7.1, 10.1, 18.4 and 4.8 μW per single unit of MHPG were achievable. The *V_OC_* and *I_SC_* values of MHPGs with varying configurations showed similar tendencies to those observed in previous studies. The voltage is generally proportional to the distance between the top and bottom electrodes and has a negative correlation with thickness [36]. Conversely, the current tends to increase linearly with the number of layers in the generator [37]. The mean electricity generation outputs for *V_OC_* values were 0.32, 0.17, 0.19 and 0.05 V, and for *I_SC_* values they were 16.57, 15.59, 47.43 and 3.02 μA from MHPG-1, MHPG-2, MHPG-3 and MHPG-4, respectively. For a comprehensive evaluation of MHPG performance, the long-term generation stability of each device was assessed at 23 °C and a relative humidity of 40% (Figure 3d–g). The generation outputs of *V_OC_* and *I_SC_* from the MHPGs quickly increased due to the maximum wetness difference between the two ends of the MHPGs. After reaching peak generation efficiency, both *V_OC_* and *I_SC_* gradually decreased and stabilized to consistent generation ranges as the MHPGs became fully wetted, maintaining a continuous water supply and evaporation. Although each MHPG exhibited different tendencies in terms of maximum electricity output, efficiency change gradient and stable output range, the results provided sufficient insight into the generation efficiency characteristics of various MHPG types for further development. The electricity generation output from MHPGs can be further enhanced by connecting the devices in series or parallel, similar to commercial batteries. For instance, in the case of MHPG-1, *V_OC_* of 1.51 V and *I_SC_* of 83.4 μA were achieved by connecting five MHPGs in series (Figure 3h) and parallel (Figure 3i), respectively.

### 3.3. Discussion and Future Perspectives

This study explored the feasibility of repurposing discarded face masks for hydrovoltaic energy harvesting, presenting a novel approach to addressing both plastic waste and energy generation challenges. The results demonstrated that face masks, modified with carbon black (CB), can effectively generate electricity through the hydrovoltaic mechanism. Following the research outcomes and findings, further implications can be discussed for both energy harvesting technology and environmental sustainability perspectives (Figure 4).

The hydrovoltaic generation output performance of the MHPGs varied with the different configurations examined. MHPG-1, which retained the original mask form, exhibited a maximum open-circuit voltage (*V_OC_*) of 0.39 V and a short-circuit current (*I_SC_*) of 18.3 μA. In contrast, MHPG-2 and MHPG-3, which involved cutting and reassembling the masks into smaller sections, showed improved performance, with maximum *I_SC_* values of 32.7 μA and 65.6 μA, respectively. MHPG-4, which involved rolling the mask, had the lowest output, suggesting that the configuration significantly affects the efficiency of the hydrovoltaic energy harvesting outputs. The variation in performance can be attributed to differences in the exposure of the mask material to the electrolyte and the evaporation surface area. MHPG-3, which had a smaller but more structured area, allowed for better exposure and interaction with the electrolyte, enhancing the overall ion transport and energy conversion efficiency. This aligns with the principle that smaller, well-defined channels can improve fluid dynamics and, consequently, the electrokinetic phenomena involved in hydrovoltaic energy generation.

The long-term stability of MHPGs was evaluated by monitoring voltage and current outputs over extended periods. The results indicated that while the initial performance was high, there was a gradual decrease in *V_OC_* and *I_SC_* as the devices became fully wetted and reached a steady-state condition. This behavior reflects the inherent limitations of hydrovoltaic energy harvesting systems, where continuous exposure to moisture and evaporation can affect the material properties and performance over time. Despite this, the MHPGs demonstrated promising potential for sustainable energy generation. The ability to convert discarded face masks into functional energy-harvesting devices addresses two critical issues: reducing plastic waste and providing a renewable energy source. By integrating these masks into energy harvesting systems, we not only mitigate the environmental impact of plastic pollution, but also contribute to the development of low-cost, eco-friendly energy solutions. However, biological and medical concerns surrounding viruses and bacteria on the discarded face masks should be further considered before integrating masks into hydrovoltaic energy harvesting systems. Thus, an affordable disinfection process is required before and after characterization of MHPGs.

This innovative use of face masks, a common and problematic waste stream, illustrates how waste-to-energy principles can be adapted to emerging technologies and materials. The repurposing of discarded face masks into MHPGs demonstrates how resource efficiency can be enhanced by using existing waste streams. Face masks, primarily composed of polypropylene, are not biodegradable and contribute significantly to plastic pollution. By transforming these masks into functional energy-harvesting devices, we effectively reduce the environmental burden of plastic waste and create a new resource for energy production. Developing hydrovoltaic energy harvesting technology using possible waste would require comprehensive consideration for economic feasibility. The cost implications of implementing CB-coated hydrovoltaic energy harvesting systems are vital for their future scalability and adoption. The use of carbon black (CB) as a coating material is relatively inexpensive, and repurposing discarded face masks as substrates further enhances cost-effectiveness. However, logistical costs associated with mask collection and preparation, as well as expenses for additives like CTAB to disperse the CB, must be considered, particularly at industrial scales. Overall, the integration of waste-to-energy principles with hydrovoltaic energy harvesting technology has several policy and practical implications. Thus, the scalability of MHPGs and their integration into existing waste management systems with affordable cost-economic perspectives should be further explored.

## 4. Conclusions

This study presents a novel approach to repurposing discarded face masks as components in hydrovoltaic energy harvesting systems. By coating face masks with carbon black (CB), we significantly enhanced their ability to absorb and transport water, resulting in improved hydrovoltaic performance. The results indicate that different configurations of the masks impact their energy output, with a maximum open-circuit voltage (*V_OC_*) of 0.39 V and a short-circuit current (*I_SC_*) of 65.6 μA. Furthermore, four different generating devices (MHPG-1, MHPG-2, MHPG-3 and MHPG-4) presented average voltages of 0.32, 0.17, 0.19 and 0.05 V, respectively, and average currents for MHPG-1 to -4 were 16.57, 15.59, 47.43, and 3.02 μA, respectively. The integration of these waste materials into energy-harvesting devices not only addresses the growing problem of municipal waste pollution, but also contributes to the development of sustainable and low-cost energy solutions. The successful application of waste face masks in this context aligns with the principles of a circular economy, offering a dual benefit of reducing environmental impact and generating renewable energy. Future research should focus on optimizing the design and scalability of these systems to maximize their practical applications and benefits. This approach represents a significant step towards innovative waste management and energy production strategies, paving the way for further advancements in sustainable technologies.

## Figures and Tables

**Figure 1 polymers-16-02515-f001:**
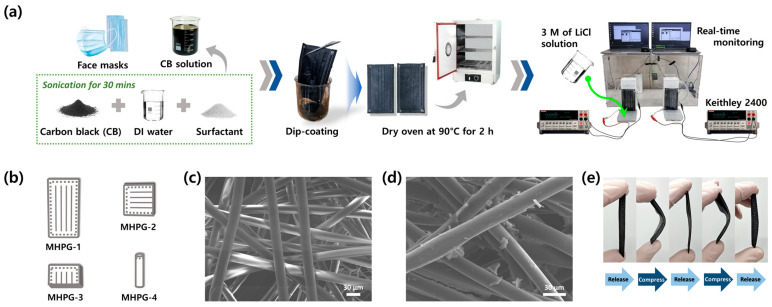
Fabrication of MHPGs and their characterizations. (**a**) Schematic diagram of MHPG preparation; (**b**) four different types of MHPG derived from a single face mask; The morphologies of (**c**) non-coated (raw) face mask and (**d**) CB-coated face mask (MHPG); (**e**) MHPG mechanical stability and flexibility tests.

**Figure 2 polymers-16-02515-f002:**
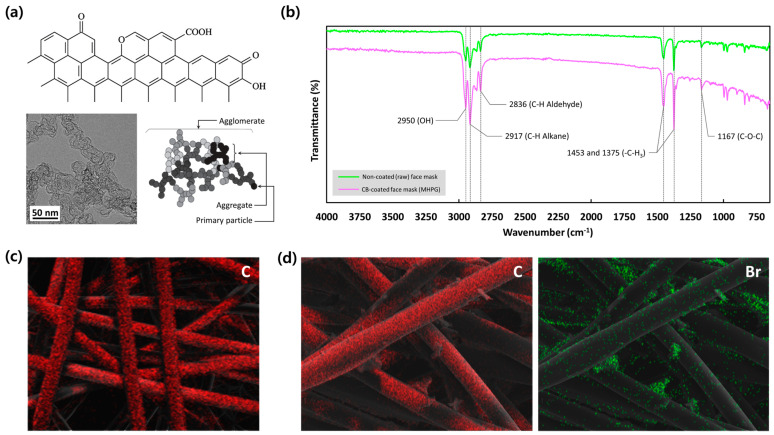
Surface modification of MHPGs. (**a**) The structure of carbon black (CB) and its agglomeration characteristics with TEM image; (**b**) FT−IR spectroscopy comparing raw and CB−coated face masks; EDS patterns of (**c**) carbon (C, red) in non-coated (raw) face mask and (**d**) carbon (C, red) and bromine (Br, green) in CB−coated face mask.

**Figure 3 polymers-16-02515-f003:**
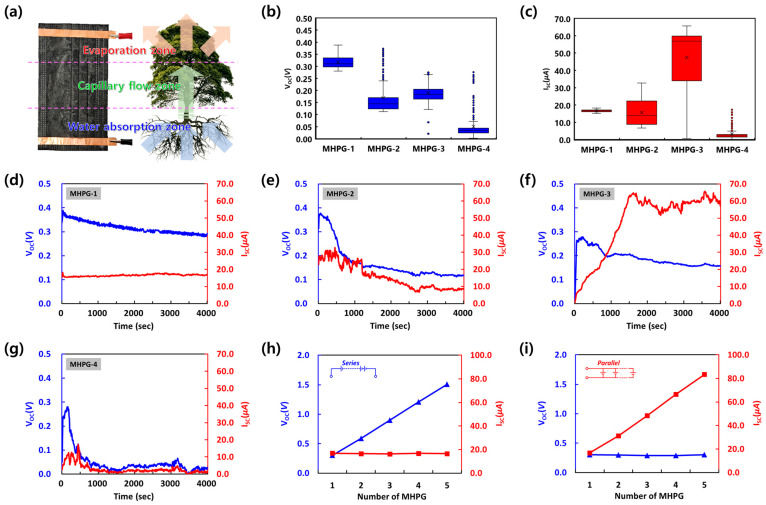
Operation principal of MHPGs and their electricity generation performance. (**a**) Schematic representation comparing MHPGs and plant transpiration; (**b**) open-circuit voltage (*V_OC_*) and (**c**) short-circuit current (*I_SC_*) distributions of different types of MHPGs. “x” and circle in those subfigures present mean values and electricity output values out of major ranges; long-term generation performance of (**d**) MHPG-1, (**e**) MHPG-2, (**f**) MHPG-3 and (**g**) MHPG-4; output voltages and currents of MHPG-1 connected (**h**) in series and (**i**) parallel.

**Figure 4 polymers-16-02515-f004:**
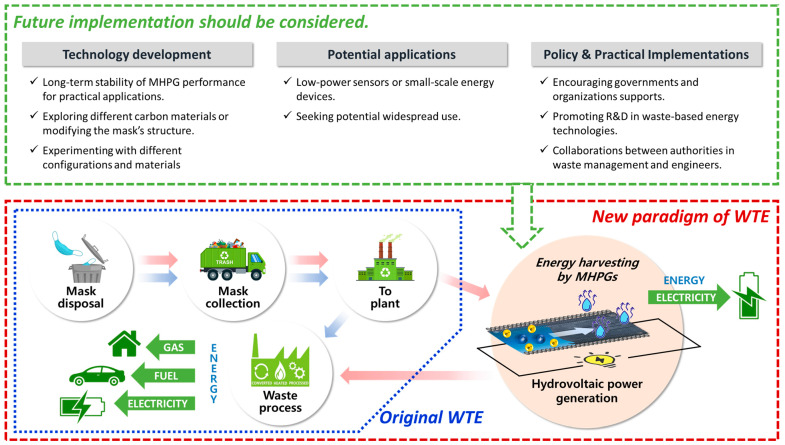
Perspectives on waste-to-energy paradigm with hydrovoltaic energy harvesting mechanism and its implications for future implementations.

**Table 1 polymers-16-02515-t001:** Different features and shapes of MHPGs derived from a piece of single-use face mask.

Types	Definitions	Sizes
MHPG-1	Original mask form	97 × 170 × 0.24 mm^3^
MHPG-2	Equally cut in 2 pieces and put together	97 × 85 × 0.48 mm^3^
MHPG-3	Equally cut in 3 pieces and put together	97 × 57 × 0.72 mm^3^
MHPG-4	Unfolded mask cut in half and rolled together	75 mm × 1.47 mm^2^

## Data Availability

The original contributions presented in the study are included in the article, further inquiries can be directed to the corresponding authors.

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
