# Peer review of "A New Paradigm on Waste-to-Energy Applying Hydrovoltaic Energy Harvesting Technology to Face Masks"

_polymers, 2024, doi:10.3390/polym16172515_

Round 1

Reviewer 1 Report

Comments and Suggestions for Authors

Kwon et al. proposed a new strategy to effectively utilize waste face masks for developing hydrovoltaic power generators. They fabricated the hydrovoltaic power generators via low-cost raw materials, carbon blacks and found that the energy output depends on the mask configurations. Addressing the environmental and economic issues caused by the wasted face masks is challenging. The reviewer supposes developing novel engineering methods to convert waste face masks into valuable devices and energy sources is very critical. As a result, I propose that this paper could be published in Polymers after addressing the minor concerns listed below.

1.     A technoeconomic analysis might be added in the introduction part, to better demonstrate the economic gains of this face-mask-based hydrovoltaic energy harvesting technology. Although this technology can alleviate environmental issues, the ultimate goal is to realize commercialization and produce economic values

2.     For further application of this technology, biological safety issues should be considered and discussed. Specifically, the virus and bacteria on the waste face mask will directly transport into water system of the hydrovoltaic technology, which might cause biological and medical concerns.

3.     For Figure 2b, the difference of FT-IR between non-coated and coated is very small, which cannot strongly support the conclusion. Hydrophilic property of MHPG can instead directly be proved by water contact angle measurement.

4.     For Figure 3a, it is great for readers to understand the mechanisms by providing this scheme. However, how can the authors claim that it is the real process happening in the hydrovoltaic device? More experimental and theoretical evidence should be offered to support this mechanism.

5.     In terms of hydrovoltaic device performance, how is this face-mask-based technology compared to other benchmark materials and systems? What is the expected energy output for this technology if it is required to replace the original waste-to-energy paradigm?

Author Response

REVIEWER #1:

Kwon et al. proposed a new strategy to effectively utilize waste face masks for developing hydrovoltaic power generators. They fabricated the hydrovoltaic power generators via low-cost raw materials, carbon blacks and found that the energy output depends on the mask configurations. Addressing the environmental and economic issues caused by the wasted face masks is challenging. The reviewer supposes developing novel engineering methods to convert waste face masks into valuable devices and energy sources is very critical. As a result, I propose that this paper could be published in Polymers after addressing the minor concerns listed below.

RESPONSE:

The authors would like to thank Reviewer #1 for the valuable comments. As suggested, we have addressed the minor concerns as outlined below.

  1. A technoeconomic analysis might be added in the introduction part, to better demonstrate the economic gains of this face-mask-based hydrovoltaic energy harvesting technology. Although this technology can alleviate environmental issues, the ultimate goal is to realize commercialization and produce economic values

RESPONSE:

We are appreciative of helpful comments from Reviewer #1. Due to the lack of related research information, the authors briefly explored about the technoeconomic analysis of different kinds of energy harvesting technologies with several former references and implicated the potential economic gains of this face-mask-based hydrovoltaic energy harvesting technology in the introduction part.

Line 89: Relatively few studies have explored the techno-economic aspects of hydrovoltaic energy harvesting systems, especially when compared to other energy conversion technologies such as triboelectric [23, 24], piezoelectric [25-27], and other renewable energy generators [28-30]. From this perspective, the potential economic benefits of hydrovoltaic energy harvesting using face masks could also be a valuable focus for future research. Finally, this aligns with the principles of a circular economy, where waste materials are reintegrated into the production cycle, minimizing waste and optimizing resource use. The significance of this research lies in its potential to provide a sustainable solution to a pressing environmental problem. By transforming discarded face masks into a source of renewable energy, this study not only addresses the issue of plastic pollution but also contributes to the diversification of energy sources.

  1. For further application of this technology, biological safety issues should be considered and discussed. Specifically, the virus and bacteria on the waste face mask will directly transport into water system of the hydrovoltaic technology, which might cause biological and medical concerns.

RESPONSE:

The authors would like to thank Reviewer #1 for the valuable suggestion. We additionally considered the biological and medical concerns by the virus and bacteria on the waste face mask.

Line 264: However, biological and medical concerns by the virus and bacteria on the waste face masks should be further considered before integrating into hydrovoltaic energy harvesting systems. Thus, affordable disinfection process is required before and after characterization of MHPGs.

  1. For Figure 2b, the difference of FT-IR between non-coated and coated is very small, which cannot strongly support the conclusion. Hydrophilic property of MHPG can instead directly be proved by water contact angle measurement.

RESPONSE:

We are appreciative of beneficial suggestion from Reviewer #1. Measuring the contact angle is well-known approach to prove the hydrophilicity of the materials, and should have been included in the manuscript for confirming the demonstrations in this study. Unfortunately, a high-speed camera for capturing the contact angle is currently unavailable due to a technical issue in our laboratory. We are seriously considering its significance for discussing the contact angle of the face mask with and without CB treatment in future studies. We once again thank Reviewer #1 for their helpful comment.

  1. For Figure 3a, it is great for readers to understand the mechanisms by providing this scheme. However, how can the authors claim that it is the real process happening in the hydrovoltaic device? More experimental and theoretical evidence should be offered to support this mechanism.

RESPONSE:

The authors would like to thank Reviewer #1 for the beneficial comment. Figure 3a illustrates the mechanism of evaporation-driven energy generation, using a natural phenomenon in plants as an example to enhance understanding of journal readers. In response to the recommendations, the authors have included further explanation about the mechanism of water evaporation-driven energy generation with several references.

Line 189: Similar to the roots of plants, the water is absorbed from one end of the MHPG follows the capillary channel. Water evaporation then occurs at the other end of the MHPG, generating electricity owing to the potential difference between the wet and dry sides (Fig. 3a). As demonstrated by Bae, et al. [34], the continuous capillary flow of injected moisture is derived at the interface between wet and dry sides of the generator. This wetting asymmetry of the device and water supply cycle are essential for generating the electric power causing streaming potential, pseudo-streaming current, electro drag effect and ion gradient diffusion, which are four principle mechanisms for hydrovoltaic conversion [31, 35].

  1. In terms of hydrovoltaic device performance, how is this face-mask-based technology compared to other benchmark materials and systems? What is the expected energy output for this technology if it is required to replace the original waste-to-energy paradigm?

RESPONSE:

We appreciate the feedback from Reviewer #1.

Various recyclable materials, such as fabric, cotton, and paper, can be incorporated into hydrovoltaic energy harvesting devices. However, in the case of MHPGs, further improvements in electricity generation output are still needed to make them viable for commercial or industrial applications, beyond conventional power banks and batteries. Many researchers have focused on developing water evaporation-driven energy generators using their own materials, such as carbon materials, ceramic membranes, cellulose, and polymers, making direct comparisons with other benchmark materials and systems challenging. Further research is needed to better understand the potential of hydrovoltaic energy harvesting technology in transforming waste into energy.

Dan Cudjoe et al (https://doi.org/10.1016/j.energy.2022.123707) briefly calculated the energy output of traditional waste-to-energy paradigms, such as incineration, and reported that the daily waste of 2.23 billion single-use face masks in Asia (equivalent to 19.12 million kg/day) could generate 32.65 million kWh/day of electricity. This equates to approximately 14.64 Wh/day per face mask. By utilizing face mask waste in a hydrovoltaic energy harvesting system before incineration, the potential electricity generation could increase to 14.64 Wh/day +α energy output.

Reviewer 2 Report

Comments and Suggestions for Authors

The comments are marked in the attached document. The authors may address the same while revising the manuscript.

Author Response

REVIEWER #2:

The comments are marked in the attached document. The authors may address the same while revising the manuscript.

RESPONSE:

The authors would like to thank Reviewer #2 for the valuable comments. As suggested, we have addressed the reviewer’s concerns as outlined below.

  1. Some more results can be presented in the abstract.

RESPONSE:

We are appreciative of helpful comments from Reviewer #2. The authors additionally presented more research outcomes in the abstract.

Line 20: The widespread use of single-use face masks during the recent epidemic has led to significant environmental challenges due to waste pollution. This study explores an innovative approach to address this issue by repurposing discarded face masks for hydrovoltaic energy harvesting. By coating the face masks with carbon black (CB) to enhance their hydrophilic properties, we developed mask-based hydrovoltaic power generators (MHPGs). These MHPGs were evaluated for their hydrovoltaic performance, revealing that different configurations and sizes of the masks affect their efficiency. The study found that MHPGs with smaller, more structured areas exhibited better energy output, with maximum open-circuit voltages (VOC) reaching up to 0.39 V and short-circuit currents (ISC) up to 65.6 μA. The integration of CB improved water absorption and transport, enhancing the hydrovoltaic performance. More specifically, MEPG-1 to MEPG-4 designated different sides and features presented mean VOC of 0.32, 0.17, 0.19 and 0.05 V as well as mean ISC of 16.57, 15.59, 47.43 and 3.02 μA, respectively. The findings highlight the feasibility of utilizing discarded masks in energy harvesting systems, offering both environmental benefits and a novel method for renewable energy generation. Therefore, this work provides a new paradigm for waste-to-energy (WTE) technologies and inspires further research into the use of unconventional waste materials for energy production.

  1. Whether mesh size of the face mask influence the interaction between water and carbon. If so, what is the optimum mesh size? How the capillarity is affected?

RESPONSE:

The authors would like to thank Reviewer #2 for the insightful observation. The size of the face mask indeed plays a crucial role in the capillarity and interaction between water and carbon black. Yulin Lv et al. (https://doi.org/10.1016/j.apenergy. 2020.115764) and Jingu Chi et al. (https://doi.org/10.1002/advs.202201586) have well demonstrated that the voltage tends to increase with the length of the generator device, while the current tends to increase with its width. In this paper, we applied four different features and shapes of MHPGs can be designated from a piece of single-use face mask. As result, MHPG-3 (97 × 57 × 0.72 mm3) showed the highest mean power output that 9.01 μW comparing MHPG-1, MHPG-2 and MHPG-4 that 5.30, 2.65 and 0.15 μW, respectively.

The width, thickness, and height of a hydrovoltaic energy harvesting device influence water flow velocity and volume, thereby affecting the output performance. Water typically evaporates primarily from the surface nanochannels of the device. The total volume flow rate is positively correlated with the device's surface area and water evaporation rate, while the overall streaming potential voltage remains nearly constant due to the presence of vertical parallel nanochannels. Increasing the thickness of the device multiplies the internal nanochannels, but it has little impact on the surface evaporation area and rate. As a result, the total volume flow rate and short-circuit current (ISC) remain almost unchanged. Since evaporation occurs mostly at the surface, there is minimal water streaming through the internal nanochannels. However, this increase in internal nanochannels leads to higher resistance and a slight decrease in output voltage. In contrast, increasing the height of the device directly enlarges the evaporating area and extends the nanochannel length, accelerating the total volume flow rate. Consequently, both the output current and voltage are significantly improved. However, if the device's height becomes too great, the limited capillary force may be insufficient to drive water flow through all the nanochannels, hindering performance.

  1. The generation of electricity with the direct interaction of water and nanocarbon structures is the underlying principle of hydrovoltaics. May be explained in detail with the relevant reactions to facilitate better understanding for the readers.

RESPONSE:

We appreciate the feedback from Reviewer #2. For better understanding for the journal readers, we additionally described more about the hydrovoltaic generation mechanisms.

Line 186: Similar to the roots of plants, the water is absorbed from one end of the MHPG follows the capillary channel. Water evaporation then occurs at the other end of the MHPG, generating electricity owing to the potential difference between the wet and dry sides (Fig. 3a). As demonstrated by Bae, et al. [34], the continuous capillary flow of injected moisture is derived at the interface between wet and dry sides of the generator. This wetting asymmetry of the device and water supply cycle are essential for generating the electric power causing streaming potential, pseudo-streaming current, electro drag effect and ion gradient diffusion, which are four principle mechanisms for hydrovoltaic conversion [31, 35].

  1. Whether aggregation shown in Figure 2 affects the efficiency of the process? Is there any method to avoid the same.

RESPONSE:

Aggregations of nano-carbon black (CB) particles are common phenomenon due to physical and chemical properties of CB. If the aggregation occurs too much, effective surface area available for interaction with water can be reduced leading to inconsistent water transport channels. This inconsistency can result in uneven water absorption and evaporation across the surface of the hydrovoltaic generator, causing fluctuations in energy output. Carbon black is conductive, and its role in hydrovoltaic systems includes facilitating charge transport. However, when particles aggregate, they may form larger clusters that can disrupt the continuity of conductive pathways, leading to lower electrical conductivity. To avoid and reduce serious aggregations of CB particles, we apply typical surfactant cetyl trimethyl ammonium bromide (CTAB). Advanced fabrication methods like electrospinning or nanopatterning can create a more organized structure that prevents particle clumping. Employing techniques such as sonication during the mixing process can also help achieve a more uniform distribution of carbon black particles, minimizing aggregation.

  1. How can CB coating gives nanostructures of carbon? Any TEM images were taken and observed.

RESPONSE:

The carbon black coating does form nanostructures due to its inherent properties and the method of dispersion used. The high surface area of these nanoparticles contributes to their unique properties, such as enhanced electrical conductivity and the ability to interact effectively with other materials like water in hydrovoltaic applications. When carbon black is dispersed in a solvent (such as deionized water), surfactants like cetyl trimethyl ammonium bromide (CTAB) can be used to prevent aggregation and maintain the nanoparticles in a well-dispersed state. This ensures that when the solution is applied to a substrate (such as a face mask), the carbon black particles remain at the nanoscale.

Furthermore, we additionally updated the TEM image of CB in Fig. 2a for mutual understanding.

Line 152: The CB-coated mask often has a porous structure due to the inherent voids and gaps between the aggregated nanoparticles. This porosity is beneficial for hydrovoltaic applications as it enhances water absorption and transport, which are critical for efficient energy harvesting. The nanostructured coating provides a large surface area relative to its volume. This high surface area is crucial for maximizing interaction with water, allowing more effective capillary action and ion transport, leading to better hydrovoltaic performance (Fig. 2a).

Line 197: Figure 2. Surface modification of the MHPG. (a) The structure of carbon black (CB) and its ag-glomeration characteristics with TEM image; (b) FT-IR spectroscopy comparing raw and CB-coated face masks; EDS patterns of (c) Carbon (C, red) in non-coated (raw) face mask and (d) Carbon (C, red) and bromine (Br, green) in CB-coated face mask.

  1. In the future perspectives, discuss the costing part of the same.

RESPONSE:

We thank Reviewer #1 for the comment. In response to the comment, the authors have included about the economic feasibility in the end of Chapter 3.3.

Line 179: Developing hydrovoltaic energy harvesting technology with possible waste would require comprehensive consideration for economic feasibility. The cost implications of implementing CB-coated hydrovoltaic energy harvesting systems are vital for their future scalability and adoption. The use of carbon black (CB) as a coating material is relatively inexpensive, and repurposing discarded face masks as substrates further enhances cost-effectiveness. However, logistical costs associated with mask collection and preparation, as well as expenses for additives like CTAB to disperse CB, must be considered, particularly at industrial scales. Overall, the integration of waste-to-energy principles with hydrovoltaic energy harvesting technology has several policy and practical implications. Thus, the scalability of MHPGs and their integration into existing waste management systems with affordable cost-economic perspectives should be further explored.
